# Stem Cells Collection and Mobilization in Adult Autologous/Allogeneic Transplantation: Critical Points and Future Challenges

**DOI:** 10.3390/cells13070586

**Published:** 2024-03-28

**Authors:** Michele Prisciandaro, Enrico Santinelli, Valeria Tomarchio, Maria Antonietta Tafuri, Cecilia Bonchi, Gloria Palazzo, Carolina Nobile, Alessandra Marinucci, Marcella Mele, Ombretta Annibali, Luigi Rigacci, Michele Vacca

**Affiliations:** 1Operative Research Unit of Transfusion Medicine and Cellular Therapy, Fondazione Policlinico Universitario Campus Bio-Medico, 00128 Roma, Italy; c.bonchi@policlinicocampus.it (C.B.); g.palazzo@policlinicocampus.it (G.P.); c.nobile@policlinicocampus.it (C.N.); a.marinucci@policlinicocampus.it (A.M.); 2Operative Research Unit of Hematology and Stem Cell Transplantation, Fondazione Policlinico Universitario Campus Bio-Medico, 00128 Roma, Italy; e.santinelli@policlinicocampus.it (E.S.); v.tomarchio@policlinicocampus.it (V.T.); m.tafuri@policlinicocampus.it (M.A.T.); marcella.mele@unicampus.it (M.M.); o.annibali@policlinicocampus.it (O.A.); luigi.rigacci@policlinicocampus.it (L.R.); 3Program in Immunology, Molecular Medicine and Applied Biotechnologies, Department of Biomedicine and Prevention, University of Rome Tor Vergata, 00133 Rome, Italy

**Keywords:** stem cell collection, mobilization, CXCR4 antagonists, autologous, allogeneic, transplant, apheresis

## Abstract

Achieving successful hematopoietic stem cell transplantation (HSCT) relies on two fundamental pillars: effective mobilization and efficient collection through apheresis to attain the optimal graft dose. These cornerstones pave the way for enhanced patient outcomes. The primary challenges encountered by the clinical unit and collection facility within a transplant program encompass augmenting mobilization efficiency to optimize the harvest of target cell populations, implementing robust monitoring and predictive strategies for mobilization, streamlining the apheresis procedure to minimize collection duration while ensuring adequate yield, prioritizing patient comfort by reducing the overall collection time, guaranteeing the quality and purity of stem cell products to optimize graft function and transplant success, and facilitating seamless coordination between diverse entities involved in the HSCT process. In this review, we aim to address key questions and provide insights into the critical aspects of mobilizing and collecting hematopoietic stem cells for transplantation purposes.

## 1. Introduction

Successful hematopoietic stem cell transplantation hinges on two critical steps: effective mobilization and efficient collection by the apheresis unit to achieve the optimal transplant dose. These key processes unlock the door to improved transplant outcomes. 

The main challenges faced by the clinical unit and collection facility in a transplant program include enhancing mobilization efficiency to optimize the collection of target components, monitoring and predicting mobilization, streamlining the apheresis procedure to reduce the number of days required for adequate collection, enhancing patient comfort by minimizing the overall collection duration, ensuring the quality and purity of stem cell products to optimize graft quality and transplantation outcome, and coordinating between the various facilities involved in stem cell transplantation.

In this review, we provide answers to questions about the crucial aspects of mobilizing and collecting hematopoietic stem cells (HSCs) for transplantation purposes.

The most common cell source for autologous and allogeneic transplantations is peripheral blood stem cells (PBSCs). As it is well known, the number of PBSCs in the peripheral blood is 0.1–0.5% [1]; therefore, it is necessary to implement mobilization strategies to increase the number and allow cell collection. What are the current mobilization strategies and future challenges?

### 1.1. Allogeneic Mobilization

Allogeneic hematopoietic stem cell transplantation (allo-HSCT), despite the introduction of several new drugs and cellular therapies that efficiently improve the outcome of high-risk hematological diseases, still represents the curative strategy for many of these life-threatening disorders [2]. 

In allo-HSCT, the three sources to collect the stem cell yield are the bone marrow (BM), the peripheral blood (PB), and the cord blood unit (CBU). 

BM was the first stem cell source adopted for allo-HSCT. BM can be collected by multiple aspirations (each one not exceeding a 5 mL volume) from the posterior superior iliac crest while the donor is under general anesthesia. The maximum volume that can be collected is 20 mL/kg of donor weight, hopefully reaching at least 3 × 10^8^ nucleated cells/kg of recipient weight [3]. 

The PB source for allo-HCT was introduced in the early 1990s as an effective stem cell source [4]. Due to the faster engraftment, the improved “graft versus leukemia” effect, the avoidance of donor exposure to general anesthesia and BM harvesting, and the favorable logistic aspects, in a few years, PB established itself as the preferred SC source for allo-HSCT [5]. 

Cord blood transplantation was first described in the 1990s [6,7] as an attractive alternative source, particularly in case of related or unrelated matched donor unavailability. The fast access to stem cell yields and the less stringent HLA-matching requirements favored the spread of CBU transplantation as a valid alternative option during the early 2000s, despite several disadvantages like the limited cell dose, the prolonged aplasia period, and the unavailability of the donor for further donations [8]. However, the introduction and the diffusion of the haploidentical donor as a valid option in case of an HLA-fully matched donor unavailability set off the near disappearance of CBU utilization [5]. 

### 1.2. Stem Cell Mobilization in Healthy Donors 

After a successful but not reproducible attempt to collect peripheral blood stem cells (PBSCs) without stimulation [9], several studies were conducted to establish the optimal mobilization strategy to adopt. Hence, the possibility of collecting an adequate stem cell yield in a suitable time through the subcutaneous administration of granulocyte colony-stimulating factor (G-CSF) alone was confirmed [4,10]. G-CSF was chosen over GM-CSF for the lower incidence of side effects [11].

G-CSF favors mobilization, promoting the degradation of adhesion molecules such as VCAM-1 that tie stem cells to the bone marrow microenvironment and by reducing the interaction between CXCR4 (a transmembrane receptor expressed on CD34+ cells) and its chemokine CXCL12, through its transcription repression, inducing peripheral blood cell migration [12,13]. Furthermore, G-CSF interacts with osteoclasts, which may promote cellular mobilization [11,14]. 

Furthermore, G-CSF induces the expansion of specific lymphocytic subsets that enrich the stem cell product. In particular, these subpopulations may provide the “graft versus host” effect, like the naïve CD4+ T cells, but may also have an immunoregulatory function associated with promoting the engraftment, like the Tregs. These concepts are the basis of graft manipulation, like T cell depletion, applied generally to avoid graft-versus-host disease. However, this topic is beyond the scope of this review, and it is deepened in specific articles [15]. 

Several experiments were initially conducted with different doses of G-CSF [16,17,18,19,20]; however, a G-CSF dose of 10 µg/kg for 5 consecutive days was associated with a better CD34+ blood peak and a higher level of CD34+ cells in the yield, compared to a G-CSF lower dose. Moreover, the peak of CD34+ level has been shown to occur 24 h following the fourth dose of G-CSF [21]. Using this evidence, the first EBMT consensus set the G-CSF dose for PBSCs mobilization in healthy donors at 10 µg/kg for 5 consecutive days; furthermore, this paper highlighted the indications to perform leukapheresis (processing up to 15 L of blood per leukapheresis by continuous flow) and the optimal CD34+ target dose to reach (2–3 × 10^6^/kg of recipient weight) [22]. 

Applying this mobilization policy, a high frequency of achieving a CD34+ level above 4 × 10^6^/kg of recipient weight through a maximum of two leukaphereses, which was associated with faster engraftment, was noted [23]. 

Some experiments applying a higher dose of G-CSF did not result in a higher CD34+ cell yield [24,25]. 

Currently, the EBMT recommends the employment of filgrastim or lenograstim 10 µg/kg/day for at least 4 days prior to stem cell collection [3]. 

Biosimilar versions of filgrastim, as the recombinant human granulocyte colony-stimulating factor (rhG-CSF), did not show any difference in mobilization in terms of efficacy and adverse events compared to the originator product; therefore, they can be safely administered for this purpose [26,27]. 

Recent studies have highlighted the feasibility of starting the stem cell collection on the 4th, instead of the 5th, day of G-CSF administration, showing no differences in terms of the CD34+ cell yield and the number of leukaphereses required to obtain the adequate cell product, but preserving donors from adverse events exposure [28,29]. However, starting the collection on the 5th day may also be appropriate, as shown both in clinical experience and experimental models [30,31]. 

The dosing schedule of G-CSF was evaluated between a single or a split daily dose to achieve a better CD34+ yield through a single leukapheresis. The split dose was assumed to be more tolerable in terms of adverse events. Although there is no clear evidence of a better result with the two daily administrations, in some cases, it proved its efficacy or at least a non-inferiority compared to the single dose [32,33,34,35]. 

Plerixafor, a CXCL12-CXCR4 antagonist, was evaluated in combination with G-CSF and alone for mobilization of hematopoietic progenitor cells (HPCs) in healthy donors. While several studies demonstrated the efficacy of the combination in achieving target CD34+ cell yields, concerns have emerged regarding the impact of plerixafor on graft composition. Specifically, concerns exist about potential alterations in immune reconstitution and associated immune processes following allogeneic hematopoietic cell transplantation (allo-HCT) due to changes in the mobilized cell population [36,37,38]. Furthermore, plerixafor alone seems to be unable to induce a successful mobilization in a significant proportion of donors [39], but a randomized trial showed that a higher dose (480 µg/kg) may ensure an adequate collection [40]. Plerixafor showed its efficacy in preventing mobilization failure in healthy, poor mobilizer donors without inducing adverse events [41,42]. Using this evidence, the Italian scientific organization GITMO introduced a specific procedure to include Plerixafor for the mobilization of healthy donors at high risk of failure [43]. 

Moreover, the G-CSF and Plerixafor combination applied as a frontline mobilization strategy appeared feasible and successful in healthy donor mobilization [44]; however, mainly due to the high rate of satisfactory grafts obtained by the G-CSF alone, this strategy cannot be pursued.

Events of unsuccessful mobilization in healthy donors are anecdotal, despite that, in these cases, especially if an alternative donor is unavailable, stem cell collection through a BM harvest could be considered. 

### 1.3. Adverse Events and Long-Term Complications 

During G-CSF stimulation, the main symptoms reported by donors are bone pain, headache, and flu-like symptoms. Bone pain and apheresis side effects are generally more frequently reported by female donors and after the first day of collection [45,46,47,48]. Despite rare cases of malignancy diagnosed during post-donation follow-up, two large prospective studies found no association between these malignancies and G-CSF stimulation, supporting the safety of G-CSF administration in healthy stem cell donors [48,49]. 

### 1.4. Autologous PBSCS Mobilization Strategies 

There are two general approaches for autologous PBSCs mobilization: steady-state mobilization using growth factors such as granulocyte colony-stimulating factor (G-CSF) alone ± CXCR4 antagonist, and chemotherapy mobilization using chemotherapy either as a part of or apart from these disease-specific treatment protocols followed by G-CSF. These strategies differ in stem cell yields, safety considerations, resource utilization, and levels of contamination of the apheresis product with tumor cells. 

Nowadays, the optimal strategy is still a matter of debate; protocols vary according to the center policy, and no definitive conclusions have emerged in recent years [50,51]. 

### 1.5. Chemotherapy Mobilization 

The choice of a chemotherapy-based mobilization regimen depends on the disease entity and institutional guidelines. 

In contrast with malignant lymphoma, where chemotherapy-based stem cell mobilization plays a significant role in disease control, the impact of chemotherapy-based stem cell mobilization seems to be negligible in MM. 

In the treatment of non-Hodgkin lymphoma (NHL), autologous stem cell transplantation (ASCT) plays a dual role: as a consolidative therapy for newly diagnosed high-risk patients and as a salvage therapy for relapsed or refractory disease. Additionally, it is employed in Hodgkin lymphoma (HL) for specific situations [52,53]. 

An ideal salvage regimen should provide sufficient disease control with acceptable hematologic and nonhematologic toxicity, and it should not negatively affect the mobilization of peripheral blood stem cells (PBSC). The mobilization and collection of adequate CD34+ PBSCs is crucial for supporting ASCT, of which G-CSF plus chemotherapy with different schemes and, in some selected cases, with the only use of G-CSF, have been the most common mobilization methods [54,55]. For patients with NHL, the infused dose of hematopoietic SCs has an important impact on engraftment kinetics; a higher mobilization target instead of MM, with an optimal dose of ≥5.0 × 10^6^/kg CD34+ HSCs, could improve engraftment and reduce complications, respiratory and infective [56,57].

For patients with MM in Europe and Western countries, a cyclophosphamide (CTX)-based mobilization strategy is widely used. Different dose levels are employed, from low dose (CTX 1.5–2 g/m^2^) to intermediate high dose (CTX 3–4 g/m^2^). 

Adequate hematopoietic progenitor cell (HPC) collection is necessary to proceed to transplantation. The target for CD34+ cell collection for a single ASCT has generally been accepted to be 3 to 6 × 10^6^ CD34+ cells/kg. Also, a dose < 2 × 10^6^ CD34+ cells/kg can have a deleterious effect on engraftment [58].

Following myelosuppressive chemotherapy, granulocyte colony-stimulating factor (G-CSF) is administered at 5–10 micrograms per kilogram of body weight (μg/kg bw) per day. Therapy typically begins 1 to 7 days after initiating chemotherapy and continues until the last day of apheresis. A new open question is about the use of novel agents such as lenalidomide (R) and anti-CD38 immunotherapy (daratumumab) during the induction phase that may impact stem-cells collections. In a recent retrospective study of 325 patients with MM who received either VTD (velcade, thalidomide, and dexamethasone) or VRD induction before ASCT, in comparison with VTD, VRD induction was associated with more frequent use of plerixafor (19.3% versus 5.4%, *p* = 0.004), which is a CXC chemokine receptor 4 (CXCR4) antagonist that improves the release of stem cells from marrow into peripheral blood [59]. In another study, the analysis of MASTER and GRIFFIN trials showed that the use of daratumumab as induction therapy in MM patients determined a 2-fold increase in the use of plerixafor [60]. In an Italian retrospective study in which patients with MM were treated with new agents (lenalidomide, carfilzomib, and daratumumab), the mobilization strategy with cyclophosphamide plus G-CSF and plerixafor “on demand” resulted in high success rate (95%) of autologous stem cell collections [61]. However, randomized clinical studies investigating standard induction triplets with or without daratumumab in MM patients showed higher use of plerixafor and lower stem cell yields in patients receiving daratumumab, regardless of the mobilization strategy adopted [62]. 

### 1.6. Steady-State Mobilization (Chemotherapy-Free) 

A steady-state mobilization with G-CSF is an effective and appealing strategy compared with a chemotherapy-based approach, particularly related to the availability of plerixafor. Retrospective and prospective studies showed the feasibility and efficacy of HSC mobilization with G-CSF-only plus ‘on-demand’ plerixafor in MM patients receiving 3–4 drugs as induction regimens [63]. 

Some studies already cited the proportion of patients with MM who successfully collected the goal target of stem cells needed to proceed to ASCT with only the use of G-CSF plus plerixafor. It was 95% in the study group of Mina et al. and 94% and 100% in the GRIFFIN and MASTER trials [60,61]. 

However, the median stem cells obtained with G-CSF in these trials and studies were lower than in the groups mobilized with chemotherapies and G-CSF; therefore, the steady-state approach should not be indicated in patients at high risk that may benefit from tandem ASCT or salvage transplant or for those who are at high risk of mobilization failure due to the presence of multiple risk factors such as bone marrow infiltration > 60% of plasma cells at diagnosis, the occurrence of grade 3–4 hematologic toxicities during induction, and lenalidomide/daratumumab-based induction regimens. 

In a study conducted in 118 European patients with hematological malignancies (90 with MM, 25 with NHL, 3 with HL), the combination of plerixafor + GCS-F was used to mobilize hematopoietic stem cells; the results showed the minimum cell yield (≥2 × 10^6^ CD34+ cells/kg) was harvested in 98% of patients with MM and in 80% of those with lymphoma in a median of one apheresis [64]. In another Asiatic study focused on 43 lymphoma patients, an acquisition success rate of frontline steady-state mobilization was 100%. The number of CD34+ cells in peripheral blood on the day before collection was a predictable index for the evaluation of stem cell collection [65]. 

Recently, a retrospective analysis conducted at a single center involving 101 individuals newly diagnosed with multiple myeloma (MM) was published. The patients underwent mobilization after treatment with lenalidomide-bortezomib-dexamethasone (RVD) and daratumumab-RVD (DRVD), with the administration of pegylated granulocyte colony-stimulating factor (G-CSF) on day 3, along with plerixafor on day 1, as a preemptive mobilization strategy. In the DRVd and RVd groups, the median number of collected CD34+ cells was 6.54 × 10^6^/kg and 6.78 × 10^6^/kg, respectively. Both groups achieved the target CD34+ stem cell collection within a median of 1 day (range: 1–4 days). However, more patients in the DRVd group required additional interventions compared to the RVd group: 51% vs. 43% needed extra plerixafor doses, and 19% vs. 14% required additional G-CSF. Notably, no mobilization failures or severe (grade 3+) mobilization-related adverse events were reported in either group. Importantly, incorporating daratumumab into the RVd induction regimen did not significantly impact stem cell yield or collection time, provided patients received preemptive G-CSF and plerixafor administration [66]. Therefore, chemotherapy-free mobilization could be an option in some cases of MM and lymphoma patients. 

### 1.7. Poor Mobilizer

Although the majority of patients are able to mobilize a sufficient quantity of CD34+ cells to collect a sufficient transplant dose with one or two collection procedures, approximately 15% fail to reach this target; these are the so-called poor mobilizers (PM) [67]. In 2012, Olivieri et al. proposed the definition of two groups of poor mobilizers: proven PM, in cases of peripheral blood CD34+ cell count never exceeding 20 cells/microliter or in cases of total collection less than 2 × 10^6^/kg in 3 procedures and predicted PM, in the presence of a previous failed mobilization, previous radiotherapy to hematopoietic sites, or previous aplastic chemotherapy. A patient who meets two of the following criteria is also a predicted PM: the presence of advanced or refractory disease, marrow involvement, severe reduction in marrow cellularity, and age over 65 years [68]. 

These two categories of patients may benefit from the use of CXCR4 antagonists.

### 1.8. Old and New Generation of CXCR4 Antagonists 

Different strategies concerning the use of plerixafor for stem cell mobilization have been adopted by different institutions, from its ‘on-demand’ or ‘just-in-time’ use (plerixafor administered according to a risk-adapted strategy based on either the number of PB CD34+ cells before the apheresis or the first CD34+ stem cell yield) to a ‘pre-emptive’ strategy in patients at high risk of stem cell mobilization failure [69]. Actually, considering the increasingly frequent use of novel agents and the diffusion of chemotherapy-free mobilization, the use of CXCR4 antagonists is in constant development; for these reasons, there are some new molecules being studied. GCP-100 is a novel CXCR4 antagonist that, in a recent in vivo study in mice, mobilized more white blood cells into peripheral blood compared to plerixafor. GPC-100-induced mobilization was further amplified by propranolol pretreatment and was comparable to mobilization by G-CSF [70]. An open-label, phase II pilot trial assessed the safety and stem cell mobilization efficacy of burixafor (GCP-100) combined with G-CSF in patients with MM, NHL, and HL. Among the nine treated patients, six (66.7%) successfully mobilized over 10 × 10^6^ CD34+ cells/kg in a single leukapheresis session. Two patients required two leukapheresis days, while one patient who had recently received lenalidomide failed mobilization initially but succeeded after a two-week recovery period. This is significantly faster than historical control patients, who typically required a median of 2–3 days of leukapheresis sessions for adequate mobilization [71]. Based on these results, an ongoing multicenter trial aimed to assess the safety and efficacy of burixafor (GPC-100) and propranolol with and without G-CSF for the mobilization of stem cells in patients with MM undergoing ASCT is now recruiting. Another molecule being studied is YF-H-2015005, a novel CXCR4 antagonist proven to increase the quantities of circulating hematopoietic stem cells in NHL. In a recent multicenter, phase III clinical trial, 101 NHL patients were randomized to receive G-CSF plus YF-H-2015005 or placebo. The primary endpoint was the proportion of NHL patients procuring ≥ 5 × 10^6^/kg CD34+ HSCs within ≤4 apheresis sessions. The proportions of patients achieving the primary endpoint were 57% and 12% in the YF-H-2015005 and placebo groups, respectively (*p* < 0.001) [72]. The GENESIS trial, a large-scale study, compared motixafortide + G-CSF to placebo + G-CSF for stem cell mobilization in multiple myeloma patients undergoing transplantation. Motixafortide significantly increased the success rate of collecting sufficient stem cells (≥6 × 10^6^ CD34+ cells/kg) within two procedures, compared to placebo (92.5% vs. 26.2%). Notably, many patients in the motixafortide group even achieved this goal in just one procedure (88.8% vs. 9.5%). Motixafortide was safe and well-tolerated, with mostly mild and temporary injection site reactions [73]. Finally, considering the more frequent use of novel agents and antibodies in MM and lymphoma, now and in the future, it is necessary to improve the efficiency of stem cell mobilization, reducing days of leukapheresis and targeting CD34+ collected with cost-effective agents. 

The GENESIS study population reported encouraging results but excluded patients with risk factors for poor mobilization. It remains to be seen whether G-CSF + motixafortide will establish itself as the new standard for mobilization in multiple myeloma or if motixafortide might replace plerixafor in routine practice for patients with poor mobilization. The data for other agents is too preliminary, and some were collected in non-oncological settings, so it is still early to understand their potential positioning in clinical practice.

Future head-to-head studies will be necessary to determine the true superiority of motixafortide over plerixafor in poorly mobilized patients.

### 1.9. Current Technologies in Stem Cell Collection

Apheresis involves three fundamental steps: collection of whole blood with the addition of an anticoagulant and subsequent separation of blood components; removal of the desired component; return of the remaining components to the donor/patient with or without replacement fluids [74]. The techniques employed by apheresis instrumentation must enable these steps to be performed efficiently, ensuring high product purity and safety for the donor/patient and the operator. What are the current technologies in use for the collection of PBSCs? 

The selection of specific cellular components for clinical transplantation via apheresis procedures has been extensively investigated over several years, aiming to establish product standardization and universally accepted quality control measures. The intrinsic variability of patients or donors, operator skill, organizational quality management systems, and interface with the processing facility all significantly impact successful collection [50]. 

In healthy individuals, the ratio of red blood cells (RBCs) to white blood cells (WBCs) is approximately 1000:1, and the ratio of platelets to WBCs is 30–60:1. As is known, PBSCs constitute only a small fraction of WBCs, typically ranging from 0.1% to 0.5% [1]. Despite mobilization strategies that elevate PBSC numbers, their isolation remains a technical challenge due to contamination from more abundant blood cells. Furthermore, the subtle differences in density between various blood cell components, including leukocyte subpopulations, coupled with the inherent heterogeneity of leukocytes, add to the complexity of the process. This poses a significant challenge for mobilization and PBSC collection, as it requires isolating a high concentration of PBSCs while minimizing contamination with other blood cells. To achieve this, all cell separators employ centrifugal force to fractionate whole blood into various components, including red blood cells, granulocytes, mononuclear cells, platelets, and plasma, based on their specific gravities. Each component can then be selectively collected using a mechanical pump. Cellular components are arranged in a stratified manner, from highest to lowest specific gravity, with PBSCs residing in the mononuclear cell layer, as shown in Figure 1. Two critical factors govern this mechanism: G-force, determined by the rotor radius, and dwell time, which represents the duration of exposure to the centrifugal field [74]. 

Some MNC collection procedures, after initial separation using centrifugal force (specific gravity), employ elutriation, a technique that utilizes two opposing forces—centrifugation and pump flow—to separate blood components based on size [74,75]. 

Cell separators employ two different flow systems to separate blood components: intermittent and continuous flow, see Table 1. The intermittent flow system extracts the desired component in cycles, emptying the separation chamber and transferring blood between the chamber and reservoir before proceeding. This cyclical approach lengthens the procedure due to repeated transfers. On the other hand, the continuous flow system extracts the desired component without interruption, maintaining a continuous blood flow. This streamlined process results in shorter procedure times and a lower extracorporeal volume compared to intermittent flow systems [74]. Extracorporeal volume (ECV) considers the total blood volume that is outside of the body, including any volume within the extracorporeal circuit. It is crucial to understand the ECV associated with cell separators and procedure protocols to determine how much blood can safely be removed from the body for instrument priming. Typically, ECV is kept below 15% of the patient’s total blood volume to minimize the risk of adverse events. This is especially important in the pediatric setting, as children have smaller blood volumes than adults [74].

Potential technical issues that may negatively impact PBSC collection include platelet loss, granulocyte content, and red blood cell content.

Although platelets and MNCs have similar specific gravities, some apheresis devices allow for the adjustment of the platelet content in the product, potentially reducing platelet loss for the patient or donor [76,77].

Another issue with PBSC collection is the presence of excessive granulocyte and RBC content.

High circulating WBC counts can adversely affect PBSC collection. Therefore, in this case, inlet flow rates should be limited to increase centrifuge dwell time to allow for adequate separation and subsequent PBSC collection [50]. 

Excessive granulocyte and RBC content typically arise from harvesting too deeply into the RBC layer. In some cases, such as low red cell MCV, it may be necessary to go deeper into the RBC layer to obtain PBSCs because low MCV is associated with poor PBSC yields, but the collection will also contain an increased number of RBC and granulocytes [78,79,80]. As suggested by Trébéden-Negre et al., an excess of granulocyte content in the collected product has been associated with delayed engraftment. They hypothesized that this detrimental effect is caused by the impairment of stem cell homing due to the pro-inflammatory cytokines and metalloproteases generated by granulocytes [81], while RBCs are susceptible to hemolysis during the freezing process. 

Therefore, it is crucial to carefully monitor the PBSC collection process and adjust the collection parameters as needed to minimize the granulocyte and RBC content in the collected product. 

Each cell separator and protocol collect PBSCs in a slightly different manner, resulting in differences in product content and in the way it is influenced by patient and donor characteristics. These differences can be utilized to tailor the collection to specific patient characteristics [50].

### 1.10. Timing and Tools for Stem Cell Collection Prediction

Effective management of stem cell collection timing is crucial for optimizing procedure scheduling within the collection facility, promoting seamless collaboration among all entities involved in a transplant program (clinical unit, collection, and processing facilities). What constitutes optimal timing for stem cell collection, and what tools are available to determine the ideal time to initiate a stem cell apheresis procedure? 

The optimal timing of hematopoietic stem cell collection is a complex decision that requires careful consideration of multiple factors, including the expertise and experience of the collection team, the patient’s unique characteristics, and the efficiency of the collection system. Considering the accreditation standards, such as JACIE-FACT, this decision is increasingly guided by a collaborative approach between clinical units and collection facilities. 

In the autologous setting with a CHT-GCSF mobilization, it is well-established that there is extreme variability among patients in reaching the peak concentration of CD34+ cells post-mobilization. All the factors that cause negative changes in the integrity of the perivascular bone marrow niches where hematopoietic stem cells reside or that always interfere negatively with chemotaxis have been extensively studied [82,83,84,85]. These factors include age > 70 years, bone marrow involvement, previous irradiation, infections, and, as recently reported, the use of daratumumab [62]. Despite this, it is common that more than 90% of patients reach the peak of hematopoietic stem cells between the 10th and 20th day of mobilization, regardless of the diagnosis and previous cycles of chemotherapy or radiotherapy [86]. However, it is also a well-established opinion that there is a huge individual difference in reaching this, for reasons that are not yet fully understood, probably of genetic origin [87,88]. In order to accurately determine the optimal timing for initiating stem cell collection, a combination of benchmarks, including best practices or consensus guidelines, should be considered [54,89]. These guidelines may recommend targeting a CD34 cell count of 20 cells/µL in mobilized peripheral blood as the starting threshold. Alternatively, time-based targets, such as initiating collection on day 11 from the start of mobilization with endonxan and GCSF, may also be employed [90]. When we introduce plerixafor into the mobilization cycle as a rescue therapy for patients who mobilize poorly after G-CSF or prophylactically when the risk of poor mobilization is high [91,92], we must ensure that the CD34 peak coincides with the collection. This introduces another variable, the timing of plerixafor infusion, which is also subject to individual variability and a highly variable growth kinetics from infusion to collection. As reported by colleagues at Mount Sinai, apheresis of 2–3 TBV (performed over 3–4 h) should be initiated 15 h after plerixafor infusion for good mobilizers and 8 h for poor mobilizers, respectively. [93]. Alternatively, as suggested by the Californian Duarte Center, which has a substantial transplant volume, apheresis should be initiated 11 h after the administration of plerixafor [94]. 

In the allogeneic setting with growth factor-only mobilization, one aspect to consider is the timing of stem cell collection—specifically, whether to initiate it on day 4 or day 5 following the initiation of mobilization therapy.

Most international organizations and registries, including the National Marrow Donor Program (NMDP), recommend day 5 for collection initiation. However, recent scientific evidence suggests that there is no significant difference in terms of the collected dose or purity of the product between day 4 and day 5 collection [29,95,96]. Moreover, donors who collect on day 4 do not appear to have an increased risk of requiring a second collection procedure compared to those who start on day 5 [97]. Finally, in the case of a need for high doses of CD34+ cells, such as in the case of haploidentical transplantation or a significant weight difference between the recipient and the donor, or if the donor is found to be a poor mobilizer, conducting the first apheresis on the 4th day would allow for immediate rescue therapy with plerixafor (between the 4th and 5th day and not between the 5th and 6th day, as reported by current studies and guidelines [29,89], thereby increasing the likelihood of achieving the target cell dose. 

All these variables suggest that it is necessary to develop additional tools to predict the optimal time for initiating stem cell collection to ensure the target dose for successful transplantation. Humpe et al. [98] investigated the decline in stem cells in the peripheral blood and the subsequent increase in the collection bag through prospective studies of kinetics between the four compartments (bone marrow, peripheral blood, cell separator, and collection bag) that make up a collection system. These findings laid the groundwork for the development of prediction algorithms. The balance between these four compartments ensures accurate prediction. The parameters that represent these four compartments and form the basis for most prediction models are identified in the hematologic and physical characteristics of the donor, the donor’s blood volume, the processed blood volume, the duration of the collection procedure, and the number of CD34+ cells present before collection. 

Several prediction algorithms have been proposed; here, we present a few examples. Delamain et al. determined the optimal timing for initiating apheresis procedures by identifying the day of the CD34 peak, calculated using the following parameters: hemoglobin concentration on the day mobilization therapy commences and the day when the CD34+ value in the peripheral blood reaches 10 cells/µL. This model proved highly effective for individuals with low mobilization potential, significantly reducing the frequency of apheresis procedures by delaying their initiation [99]. Pierelli et al. published an algorithm based on a multicenter prospective study [100]. The model employs simple parameters, some of which are known prior to collection, such as the donor’s weight or the CD34+ cell count in the peripheral blood prior to collection and the transplant dose required. Other parameters are predicted, such as the blood volume to be processed. The algorithm relies on the efficiency of the apheresis procedure. This is a crucial aspect, as no model can function effectively without knowledge of the collection system’s efficiency. The high correlation between the predicted and actual data confirms the algorithm’s accuracy. 

The application of predictive algorithms provides a valuable tool to aid decision-making in various aspects of stem cell collection, including the timing of collection, procedure scheduling, processing requirements, the number of required procedures, procedure duration, and minimizing the storage of surplus product.

### 1.11. Management Strategies for Intra- and Peri-Procedural Adverse Events in Stem Cell Collection

Collection Units prioritize patient and donor safety. What are the key principles and management strategies for intra- and peri-procedural adverse events (AEs)? 

Apheresis procedures are generally well-tolerated for both therapeutic and collection purposes. However, the potential for adverse events, expected and unexpected, exists. These side effects are categorized by severity as mild (tolerated without medications), moderate (need for medication), severe (interruption due to the AE), or death (due to AE).

The 2023 update of the World Apheresis Association (WAA) apheresis registry analyzed data from 58,355 procedures performed on 9500 patients, examining both therapeutic and collection purposes. During the period 2018–2022, 3% of procedures reported adverse events (AEs), which included technical difficulties and problems with vascular access. Among these AEs, the most frequent were mild (1.5%) and moderate (1.4%). Tingling, pricking sensations, hives (urticaria), and low blood pressure (hypotension) were the main types of mild and moderate AEs. Severe AEs were very rare, occurring in only 0.15% of procedures. Excluding access and technical issues, the update reported the following breakdown by severity. Mild AEs were 1.5% (hypotension 13.2%, tingling/pricking 38.2%, nausea/vomiting 10.3%. Moderate AEs were 1.4% (tingling/pricking 74.3%, urticaria 8.4%, hypotension 5.2%, nausea 3.9%. Severe AEs were 0.1% (syncope/hypotension 25.4%, urticaria 25.4%, arrhythmia/asystole 4.8%, nausea/vomiting 3.2%, chills/fever 1.6%) [101]. 

Considering the above, PBSC collection by apheresis can be performed safely in an outpatient setting. 

One of the most frequent PBSC adverse events is citrate-related hypocalcemia. 

The preferred anticoagulant for PBSC is a citrate solution usually containing sodium citrate dihydrate 22 g/L, glucose monohydrate 24.5 g/L, and citrate acid monohydrate 8 g/L, namely, ACD-A—Anticoagulant Citrate Dextrose Solution A [102]. 

Citrate exerts its action through the chelation of divalent cations, primarily calcium and magnesium. This chelation, occurring during apheresis due to citrate infusion, can lead to reductions in serum concentrations of ionized calcium and magnesium. Ionized calcium levels can decrease by 25% or more during an apheresis procedure [50]. Mild reactions to citrate chelation during apheresis may manifest as metallic taste, perioral, and/or acral paresthesia. Moderate reactions are characterized by the persistence of these symptoms despite supportive measures, such as reduced blood flow rate, increased anticoagulant-to-whole blood ratio, or calcium supplementation. Additionally, nausea, vomiting, abdominal pain, shivering, lightheadedness, tremors, and hypotension reminiscent of hypovolemia or vasovagal reactions may occur. In severe cases, symptoms can escalate to carpopedal spasm, tetanic seizures, and cardiac arrhythmias, particularly QT prolongation. However, hypocalcemia is usually mild and easily managed with oral or intravenous calcium supplements. 

Citrate-based anticoagulation also reduces plasma potassium and free magnesium levels. Patients undergoing PBSC collection should have an electrolyte panel drawn within 24–74 h of the procedure, and low serum potassium and magnesium levels should be replaced pre-procedure with oral or IV electrolyte supplements [74].

During apheresis procedures, regardless of the device or collection program employed, careful consideration must be given to the extracorporeal blood volume (ECV) in low-weight patients to mitigate potential hemodynamic alterations. These alterations can manifest as hypotension or vasovagal reactions.

Two distinct mechanisms can lead to hypotension, requiring different management approaches.

Hypotension due to intravascular volume depletion typically presents accompanied by tachycardia and tachypnea. To address this, the apheresis procedure is usually paused, and the patient receives a fluid bolus followed by evaluation by a physician. 

Hypotension due to vasovagal reaction exhibits concurrent bradycardia, differentiating it from hypotension due to hypovolemia. This reflex response of the parasympathetic nervous system leads to vasodilation, causing significant hypotension (blood pressure as low as 50/20 mmHg) without the reflex tachycardia typically seen in hypovolemia. This is a key distinction between the two conditions.

The overactive parasympathetic response can be triggered by anxiety, pain, or hypocalcemia during the procedure. Management involves stopping the apheresis and implementing supportive measures to improve patient comfort and hemodynamics.

Beyond vital sign changes, vasovagal reactions can manifest with pallor, diaphoresis, nausea and vomiting, syncope, potential convulsions, and urinary or fecal incontinence.

In some cases, this presentation can closely resemble epileptic seizures, highlighting the importance of accurate diagnosis. 

With good surveillance, it is often possible to prevent a vasovagal reaction with distraction techniques. Engaging the patient in conversation or other distracting activities often results in a return to the baseline state [74]. Upon the onset of a vasovagal reaction during apheresis, initial management focuses on alleviating patient distress and promoting hemodynamic stability. This often involves implementing non-pharmacological interventions such as deep breathing exercises, coughing, laughing, or repositioning the patient to reduce discomfort and enhance comfort. These measures can effectively resolve mild vasovagal reactions.

However, in more severe cases, escalation of therapy may be necessary. This may involve procedural intervention, including a temporary pause in donors or patients. Additionally, Trendelenburg positioning, which elevates the lower extremities to facilitate venous return and improve blood flow to the heart, can be implemented. Furthermore, fluid resuscitation through the administration of a fluid bolus can help restore intravascular volume and blood pressure. 

Additional adverse reactions are associated with the type of venous access used for apheresis procedures. 

Peripheral veins are the preferred choice for cell collection. However, between 0.6% and 20% of donors, depending on the apheresis center, may not have suitable peripheral veins to achieve the necessary blood flow for collecting PBSCs [49,103,104,105,106]. For autologous patients, if venous access is deemed questionable, a CVC is the predominant type of vascular access used to harvest HSCs. Current policies and practices regarding the assessment, placement, and management of venous access for HPC collections for healthy donors are based on institutional preferences and/or National Health Service-mandated policies. 

A number of these regulatory agencies recommend the use of peripheral venous access for healthy donors, especially unrelated donors. 

If peripheral venous access is unavailable, the use of a CVC should be considered based on appropriate clinical indication and it shall be placed by qualified personnel. 

However, CVCs should be a last resort for healthy stem cell donors. Peripheral access should always be prioritized whenever possible to ensure donor safety. Bone marrow harvest can be considered as an alternative option. [107,108].

Another option is the use of midlines, which have been shown to be effective in collecting HSCs in adult donors who do not have suitable peripheral vascular access [43,109,110]. 

It is important to note that both peripheral and central venous access methods carry a potential risk of AEs.

Peripheral access may be associated with bruising, hematomas, nerve injuries, infections, phlebitis, and/or deep vein thrombosis. CVCs can be associated with several potential complications, including infection, thrombosis, hemorrhage, air embolism, pneumothorax, hemothorax, and arrhythmias.

## 2. Conclusions

Continuous optimization of stem cell mobilization and collection protocols can lead to a substantial improvement in this therapeutic approach, particularly for allogeneic stem cell transplantation. Increased efficiency, streamlined patient scheduling and processing, and prioritized patient comfort will not only enhance graft quality and ensure transplant success but also facilitate broader patient access and improve quality of life. 

In addition to the described scenarios concerning stem cell collection, novel challenges are emerging with respect to the collection of cellular effectors, such as CAR T cells and other immunotherapeutic agents. These challenges are related to the production of biological drugs and gene editing, further emphasizing the critical importance of product quality, including high purity, viability, and specific functionalities. Nevertheless, these new scenarios further solidify the leading role of the collection unit in the field of precision medicine.

## Figures and Tables

**Figure 1 cells-13-00586-f001:**
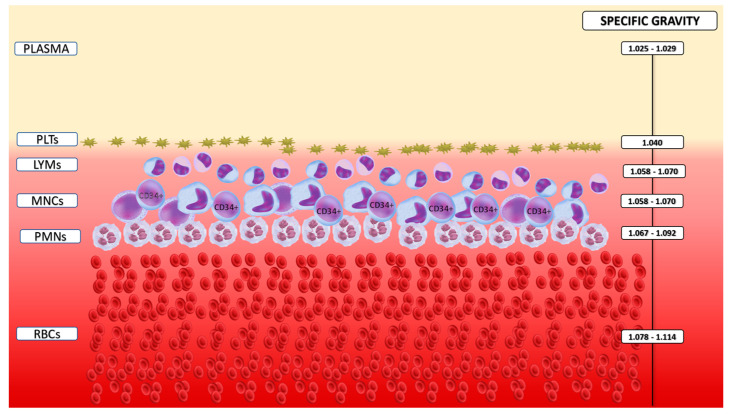
Specific gravity of blood cells and plasma. PLTs, Platelets; LYMs, Lymphocytes, MNCs, Mononuclear cells; CD34+, Hematopoietic stem cells; RBCs, Red blood cells.

**Table 1 cells-13-00586-t001:** MNC collection instruments.

MNC COLLECTION INSTRUMENTS	TYPE OF SYSTEM	NEEDLE SYSTEM (ECV Approx. [50,74])	KIT AND PROTOCOL
FRESENIUS-KABI			
AMICUS^®^	CFC	Double needle (163 mL)	MNC Kit (works in cycles, centrifuge)
COM.TEC^®^	CFC	Double needle P1YA, P1Y, (177 mL) * C4Y, RVY (193 mL) *	P1YA, P1Y, C4Y, RVY (works in cycles, centrifuge)
HAEMONETICS CORP			
MCS+^®^ 9000	IFC	Single needle (Variable 125 bowl 380 (38% HCT)-259 (52% HCT)/82–87 mL) *	PROTOCOL PBSC 0971E-00 (works in cycles, centrifuge)
TERUMO BCT INC			
SPECTRA OPTIA^®^	CFC	Double needle MNC (191 mL) CMNC (297 mL)	MNC protocol (works in cycles, centrifuge + elutriation) CMNC protocol (centrifuge)

ECV approx.: extracorporeal volume approximately; IFC = intermittent flow centrifugation; CFC = continuous flow centrifugation; * User manual.

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
