# Peer review of "Stem Cells Collection and Mobilization in Adult Autologous/Allogeneic Transplantation: Critical Points and Future Challenges"

_cells, 2024, doi:10.3390/cells13070586_

Round 1

Reviewer 1 Report

Comments and Suggestions for Authors

The present review aims to provide insights into the critical aspects of hematopoietic stem cell mobilization and collection that are the main challenges in transplantation. Very interesting and innovative topic.

The references are in accordance with the cited texts.

The figure is useful and nice.

I report the corrections I deem necessary:

Abstract: from line 21 there is a list, but each sentence is separated like if they were all separate periods. Rewrite

Introduction. Line 45-49: Remove bold.

Lin 47: where is the reference for this information “the number of PBSCs in the 46 peripheral blood is 0.1-0.5%”?

Line 61: it is probably 3 x 108 and not 3x108. Correct.

Line 80: Specify the mode, route of administration of granulocyte 80 colony-stimulating factor (G-CSF)

 Line 120: “Plerixafor, an antagonist of CXCL12-CXCR4 interaction was tested with G-CSF and alone for healthy donor mobilization.”  Write more clearly the concept of single or combined administration

Lines 134-137: Unclear period. Rephrase

Line 160: “HematopoieticSCs” space

Lines 235-237: Unclear period. Rephrase

Line 275-281: Remove the bold and rework without the question.

347-351: Remove bold. The same applies to subsequent periods

Author Response

Thank you very much for taking the time to review this manuscript. 

3. Point-by-point response to Comments and Suggestions for Authors 

The present review aims to provide insights into the critical aspects of hematopoietic stem cell mobilization and collection that are the main challenges in transplantation. Very interesting and innovative topic. 

The references are in accordance with the cited texts. 

The figure is useful and nice. 

I report the corrections I deem necessary: 

Abstract: from line 21 there is a list, but each sentence is separated like if they were all separate periods. Rewrite.  

Introduction. Line 45-49: Remove bold.  

Lin 47: where is the reference for this information “the number of PBSCs in the 46 peripheral blood is 0.1-0.5%”?  

Line 61: it is probably 3 x 108 and not 3x108. Correct.  

Line 80: Specify the mode, route of administration of granulocyte 80 colony-stimulating factor (G-CSF).  

 Line 120: “Plerixafor, an antagonist of CXCL12-CXCR4 interaction was tested with G-CSF and alone for healthy donor mobilization.”  Write more clearly the concept of single or combined administration.  

Lines 134-137: Unclear period. Rephrase  

Line 160: “HematopoieticSCs” space.  

Lines 235-237: Unclear period. Rephrase  

Line 275-281: Remove the bold and rework without the question.  

347-351: Remove bold. The same applies to subsequent periods  

Response 1 

Abstract: from line 21 there is a list, but each sentence is separated like if they were all separate periods. Rewrite. We rewrote. 

Introduction. Line 45-49: Remove bold. The paper is structured like an interview. Any questions are written in bold. We leave the choice to remove bold to the Editor 

Lin 47: where is the reference for this information “the number of PBSCs in the 46 peripheral blood is 0.1-0.5%”? We added it. 

Line 61: it is probably 3 x 108 and not 3x108. Correct. We correct it 

Line 80: Specify the mode, route of administration of granulocyte 80 colony-stimulating factor (G-CSF). We added route of administration. 

 Line 120: “Plerixafor, an antagonist of CXCL12-CXCR4 interaction was tested with G-CSF and alone for healthy donor mobilization.”  Write more clearly the concept of single or combined administration. The concept was rephrased. 

Lines 134-137: Unclear period. Rephrase The concept was rephrased. 

Line 160: “HematopoieticSCs” space. We correct it 

Lines 235-237: Unclear period. Rephrase The concept was rephrased. 

Line 275-281: Remove the bold and rework without the question. The paper is structured like an interview. Any questions are written in bold. We leave the choice to remove bold to the Editor 

347-351: Remove bold. The same applies to subsequent periods The paper is structured like an interview. Any questions are written in bold. We leave the choice to remove bold to the Editor 

Reviewer 2 Report

Comments and Suggestions for Authors As authors claimed in the abstract section, the present manuscript is aimed to address
key questions and provide insights into the critical aspects of mobilizing and collectionhematopoietic stem cells for transplantation purposes. However, the main problem withthe manuscript is that many HSC mobilization and collection current key points are
neither mentioned nor discussed. For instance, Pediatric issues ( either autologous and allogeneic) Minors as transplant donors, Defining bad mobilizers and how to deal with them, Mobilization regimens according to diseases ( pediatric solid tumors, lymphoma and or myeloma patients) Cell targets for transplantation among others.
I think that the manuscript should be scheduled in autologous (first) and allogeneic
issues and should include specific comments for pediatrics and adult patients.
Comments on the Quality of English Language

Just minor typo errors

Author Response

Thank you very much for taking the time to review this manuscript. The feedback was greatly appreciated and contributed to improving the clarity and comprehensibility of the article. 

Round 2

Reviewer 2 Report

Comments and Suggestions for Authors

I think that authors have adequately addressed the suggestions and criticisms

Comments on the Quality of English Language

I have no comments 

Author Response

Thank you very much for taking the time to review this manuscript.